# Dynamic HIV-1 spike motion creates vulnerability for its membrane-bound tripod to antibody attack

Shuang Yang[1,9], Giorgos Hiotis[1,2,9], Yi Wang[3,4], Junjian Chen[3,4], Jia-huai Wang[3,5,6,7], Mikyung Kim[3,8], Ellis L. Reinherz [ID][3,4] ✉ & Thomas Walz [ID][1] ✉

Vaccines targeting HIV-1's gp160 spike protein are stymied by high viral mutation rates and structural chicanery. gp160's membrane-proximal external region (MPER) is the target of naturally arising broadly neutralizing antibodies (bnAbs), yet MPER-based vaccines fail to generate bnAbs. Here, nanodisc-embedded spike protein was investigated by cryo-electron microscopy and molecular-dynamics simulations, revealing spontaneous ectodomain tilting that creates vulnerability for HIV-1. While each MPER protomer radiates centrally towards the three-fold axis contributing to a membrane-associated tripod structure that is occluded in the upright spike, tilting provides access to the opposing MPER. Structures of spike proteins with bound 4E10 bnAb Fabs reveal that the antibody binds exposed MPER, thereby altering MPER dynamics, modifying average ectodomain tilt, and imposing strain on the viral membrane and the spike's transmembrane segments, resulting in the abrogation of membrane fusion and informing future vaccine development.

The genesis of human immunodeficiency virus-1 (HIV-1) as a pathogen in *Homo sapiens* is thought to have occurred in the Democratic Republic of Congo around 1920 consequent to the retrovirus jumping from chimpanzee to human, followed by initiation of the current epidemic in the 1970's[1,2]. Attesting to the importance of this zoonosis, data compiled in the last 35 years indicate that ~78,000,000 individuals globally have been infected and ~35,000,000 have died despite multi-agent, anti-viral drug therapy[3].

The trimeric gp160 HIV-1 envelope spike protein (Env), a transmembrane glycoprotein comprising three protomers each of gp120 and gp41, is the singular virus-derived protein on the virion. Hence, it is the sole target of protective antibodies arising naturally in patients or elicitable in uninfected individuals through vaccination[4–6]. That said, vaccine induction of antibodies with requisite breadth against diverse viral strains, so-called broadly neutralizing antibodies (bnAbs), has

failed to engender antibodies that either block the initial virus binding to human CD4 T cells or that inhibit subsequent Env conformational changes and attendant fusion events required for post-binding viral entry into the host cell (reviewed in ref. 7). Immune recognition of Env is hampered by its extraordinary sequence variability[8], dense glycosylation[9,10], and conformational masking of key sites and metastable state[11]. Nonetheless, some chronically HIV-1-infected patients develop bnAbs over years of infection[4–6]. Attendant viral diversification and extensive antibody somatic mutations improve binding affinity, optimizing the match between paratopes and epitopes[12]. BnAbs are directed against the CD4-binding site as well as the V1V2 and V3 glycan sites on gp120, the gp41-gp120 interface region and the gp41 membrane-proximal external region (MPER)[4–6]. The MPER, which connects the Env ectodomain to its transmembrane (TM) domain, is amongst the most conserved HIV-1 segments across clade strains[13].

[1]Laboratory of Molecular Electron Microscopy, The Rockefeller University, New York, NY, USA. [2]Tri-Institutional PhD Program in Chemical Biology, The Rockefeller University, New York, NY, USA. [3]Laboratory of Immunobiology, Department of Medical Oncology, Dana-Farber Cancer Institute, Boston, MA, USA. [4]Department of Medicine, Harvard Medical School, Boston, MA, USA. [5]Department of Biological Chemistry and Molecular Pharmacology, Harvard Medical School, Boston, MA, USA. [6]Department of Cancer Biology, Dana-Farber Cancer Institute, Boston, MA, USA. [7]Department of Pediatrics, Harvard Medical School, Boston, MA, USA. [8]Department of Dermatology, Harvard Medical School, Boston, MA, USA. [9]These authors contributed equally: Shuang Yang, Giorgos Hiotis. ✉e-mail: ellis_reinherz@dfci.harvard.edu; twalz@rockefeller.edu

That conservation and the observation that naturally arising MPER-targeting bnAbs manifest the broadest neutralization breadth have nominated the MPER as a key target for vaccine design.

The structure of the MPER by itself and together with the TM domain in a membrane-mimetic environment has been studied extensively by spectroscopic methods, establishing that the MPER peptide tends to adopt a membrane-embedded kinked helix-hinge-helix conformation with additional flanking flexibility[13,14] and suggesting that bnAb-bound MPER is partially extracted from the membrane[15]. However, MPER peptide vaccines have failed to elicit bnAbs and the exact disposition of the MPER as part of the entire membrane-embedded Env trimer remains controversial. One cryo-electron tomography (cryo-ET) study reported that the MPERs form a tripod-like structure composed of three separated helices[16], but other cryo-ET studies showed that the three MPER segments are organized into a compact stalk[17,18]. Therefore, for rational vaccine design purposes, a high-resolution structure is needed to elucidate the actual disposition of the MPER in the context of both a lipid bilayer and the complete and glycosylated ectodomain of the Env trimer.

Here, we reconstitute mammalian cell-expressed HIV-1 Env protein into lipid nanodiscs and use single-particle cryo-electron microscopy (cryo-EM) to determine its structure alone as well as in complex with up to three Fabs of the bnAb 4E10. Together with coarse-grained molecular-dynamics (CGMD) simulations, these structures reveal the dynamics of the Env ectodomain and the MPER on the membrane. In addition, the conformational changes in the MPER along with inferred perturbations in vicinal lipids and TM domains that accompany the binding of increasing numbers of these 4E10 Fabs are ascertained.

## Results

### The MPER segments of membrane-embedded Env protein form a tripod

The gp145 SOSIP construct[19] of the Env spike protein from the BG505 clade A strain, consisting of the entire ectodomain (residues 1–662), the MPER and transmembrane segments and 17 residues of the cytoplasmic domain (Fig. 1a), was expressed in HEK293 cells. The C-terminally truncated Env construct was used due to its much better expression compared to full-length protein. After purification in dodecyl maltoside (DDM) using the PG9 Fab for affinity chromatography followed by size-exclusion chromatography (SEC) (Supplementary Fig. 1a, b), the recombinant gp145 was reconstituted into nanodiscs with a molar 1.5:1:1.07 mixture of palmitoyl oleyl phosphatidylcholine (POPC), palmitoyl oleyl phosphatidylglycerol (POPG) and brain polar lipid extract. This lipid mixture is similar in composition to that of the HIV-1 membrane[20], except that it is missing cholesterol. Even though cholesterol is the most abundant component of the HIV-1 lipidome, our CGMD simulations performed with a membrane containing 20% cholesterol exhibited dynamic behavior of the ectodomain, in particular tilt angles, consistent with those seen in our cryo-EM maps (see below). Empty nanodiscs were removed by SEC (Supplementary Fig. 1a). SDS-PAGE analysis of the SEC peak fraction confirmed the presence of both gp145 and membrane scaffold protein MSP1D1dH5, which was used for nanodisc assembly (Supplementary Fig. 1b). EM images of negatively stained gp145 in DDM showed mostly three-fold symmetric top views of gp145 (Supplementary Fig. 1c), whereas negative-stain EM images of nanodisc-embedded gp145 showed side views and revealed the nanodiscs (Supplementary Fig. 1d). Nanodisc-embedded gp145 was then incubated with the Fab fragment of bnAb 4E10, cross-linked with BS3 and vitrified on grids. Image processing with RELION-3 showed that ~80% of the selected particles had no obvious density for the 4E10 Fab. Further processing of these particles yielded a density map of nanodisc-embedded gp145 at an overall resolution of 3.9 Å (Fig. 1b and Supplementary Figs. 2–4). This map allowed us to build an atomic model for ectodomain residues 31–662 and a backbone model for residues

663–671 (see "Methods" for more detail). The map did not resolve the transmembrane helices, which were also not resolved in a previous map of nanodisc-embedded full-length Env protein[21], most likely the result of variations in the position of the single-span TM domains relative to the nanodisc. However, additional low-resolution density (-5 Å; Supplementary Fig. 4b) represented the beginning of the MPER-N segment, into which we placed the N-terminal segment of an NMR structure of the MPER (PDB: 2PV6)[14] (Supplementary Fig. 4d), thus revealing the disposition of the MPER in the context of a membrane and the trimeric Env ectodomain. The MPER-N segments of each protomer form three independent helices that adopt a tripod-like conformation (Fig. 1b, c), which differs from the previously reported structures in which the MPERs either converge[16] or form a compact stalk[17,18].

Our structure indicates that the MPER-N segment is partially buried in the lipid bilayer (Fig. 1b), consistent with a previous NMR structure that showed the MPER to form membrane-immersed helical segments[14]. Due to a hinge connecting the MPER-N with the MPER-C helices[14] and the lack of reliable density for the MPER-C segment and the TM region, our map does not inform on the position of the MPER-C segments or the organization of the TM helices. That said, as discussed below, it is likely that each MPER-C segment extends back toward the center of the Env trimer and that the MPER-linked TM domains of each protomer approximate near the three-fold trimer axis in either a loose association or, alternatively, one related to the TM bundle observed in two NMR structures of MPER-TM[22,23]. Of note, other studies imply that the TM helices in the native trimer spike do not form a rigid trimer[24] but might transition into such a configuration during the viral fusion process. In one of the NMR structures (PDB: 6E8W)[22], the MPER-C segments form somewhat continuous helices with their respective TM domains, albeit the MPER-C helices are distorted and bend away from the three-fold axis. The hinge region between the MPER-C and -N segments then form strong kinks, resulting in the MPER-N segments approximating at the three-fold axis. We will refer to this as the stalk–bubble conformation. In the other NMR structure (PDB: 6DLN)[23], it is the hinges between the TMD and MPER-C that form strong kinks whereas the MPER-C and -N segments form a continuous helix that extends away from the three-fold helix, forming a tripod conformation.

To compare the conformation of the MPER in the context of the full Env trimer, we measured the distance between the N-terminal end of the MPER segments, i.e., the Cα atom of residue Lys665, between adjacent protomers. In our cryo-EM structure, the distance is 30 Å, while the distance is 53 Å in the NMR tripod structure and 16 Å in the NMR stalk–bubble structure (Supplementary Fig. 5a). The measurement established that the MPER segments in our structure adopt a conformation in between the two NMR structures. However, the Trp666 residues from the three MPER-N protomers converge in the stalk–bubble conformation to form a hydrophobic core, which is not the case in the cryo-EM structure (Supplementary Fig. 5b). Furthermore, placing the two NMR structures into the cryo-EM map, using the MPER-N segments as anchor points, the dimension of the NMR tripod structure fits much better into the map, establishing that this structure is closer to the MPER conformation in the context of the entire Env protein (Fig. 1d). The remaining difference in MPER conformation between the NMR tripod structure and our cryo-EM structure is most likely due to constraints imposed on the MPER by the ectodomain, which was present in the cryo-EM study but missing in the NMR study.

### The Env protein undergoes spontaneous tilting motions

Data processing of nanodisc-embedded gp145 revealed that the ectodomain can adopt a wide range of angles with respect to the membrane plane. To better characterize the range of ectodomain tilting, we performed iterative 3D classifications (Supplementary Fig. 3), resulting in 20 maps with well-defined density for the nanodisc

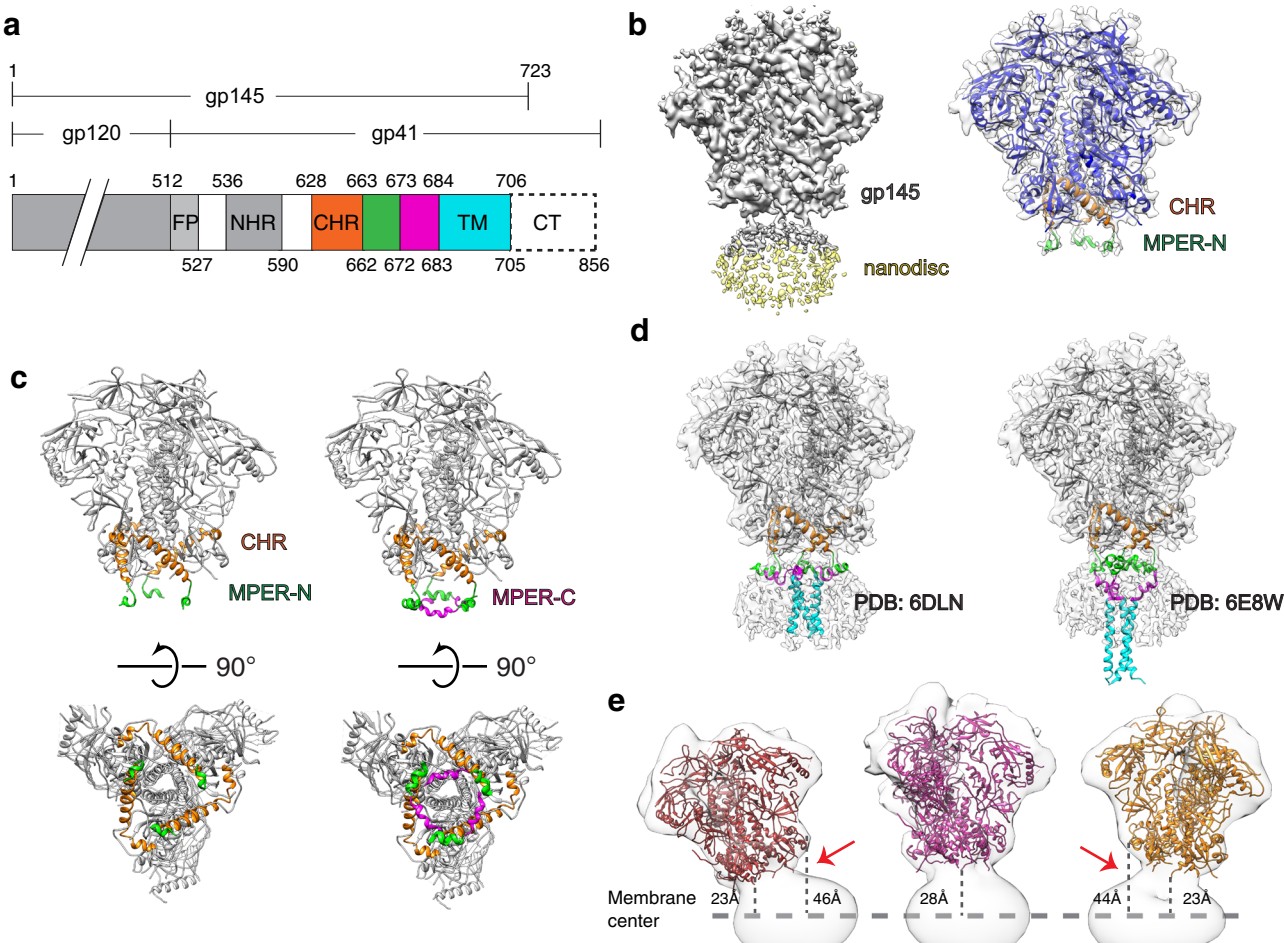

**Fig. 1 | Cryo-EM structure of nanodisc-embedded HIV-1 gp145. a** Domain architecture of gp160. FP, fusion peptide; NHR, N-terminal heptad repeat; CHR, C-terminal heptad repeat (orange); MPER-N (green); MPER-C (magenta); TM, transmembrane domain; CT, cytoplasmic tail. Numbers above indicate the first residues of the segments and the numbers below the last. **b** Left: Cryo-EM map of nanodisc-embedded gp145 at a low contour level (0.011). The density for the nanodisc is shown in light yellow. Right: Cryo-EM map of nanodisc-embedded gp145 at a higher contour level (0.014) shown in semi-transparent gray, with the modeled gp145 structure, shown in ribbon representation, fit into the map. The CHR region is shown in orange and the MPER-N segment in green. The rest of the ectodomain is shown in blue. **c** Left: Structure of gp145 seen parallel (top) and perpendicular to the membrane plane (bottom). Right: Same views as in the left panel after adding the NMR structure of the complete MPER (PDB: 2PV6) to our gp145 model (based on an overlay of the MPER-N segments). The CHR region is shown in orange, the MPER-N segment in green, and the MPER-C segment in magenta. **d** Placement of the two available NMR structures of MPER-TM into the cryo-EM map (placed based on the position of the MPER-N segments). The tripod structure (PDB: 6DLN) fits well into the map (left), whereas the MPER-N segments of the stalk–bubble structure (PDB: 6E8W) correspond poorly with the cryo-EM map as the transmembrane helices protrude far from the nanodisc density (right). **e** Three cryo-EM maps of nanodisc-embedded gp145 showing that the gp145 ectodomain adopts a wide range of angles with respect to the membrane plane. The maps are shown as semi-transparent surfaces, and the structure of the ectodomain as colored ribbons. Distances were measured from the end of the ectodomain (residue Ala662) to the center of the nanodisc (indicated by the dashed line).

and sufficient features in the ectodomain density to unambiguously dock the atomic model (examples shown in Fig. 1e). A movie generated from the docked models illustrates the degree of tilting the Env trimer can undergo relative to the nanodisc membrane (Supplementary Movie 1). The maps show that the three-fold axis of the ectodomain can tilt up to at least 30° from the membrane normal (Fig. 1e and Supplementary Movie 2). As a result of the tilt, the distance of the ectodomain from the surface of the membrane increases by as much as 20 Å (Fig. 1e).

To better understand the dynamics of the ectodomain on the membrane, we used CGMD simulations (Fig. 2a). gp145 was embedded into a POPC bilayer containing 20% cholesterol, since the HIV-1 membrane is known to be rich in cholesterol[20]. The simulations corroborated the cryo-EM results in that the ectodomain is highly mobile (Supplementary Movie 3) and can tilt as much as ~63°. From five 10-μs simulations, the ectodomain is on average tilted by 17.6° ± 10.0° but assumes a wide range of tilts (Fig. 2b). The CGMD simulations also

show the MPERs to be highly dynamic and to occasionally leave the membrane plane (Supplementary Movie 3), but there are no indications that the MPER movements are correlated with the tilting of the ectodomain. Conversion of CGMD snapshots to atomistic views also showed that high ectodomain tilts per se did not favor extraction of the MPER from the membrane (Fig. 2c).

**bnAb 4E10 binds to a tilted Env protein and exerts force on the TM region**

The CGMD results indicated that MPER-specific bnAb binding to gp145 depends on two independent dynamics, (1) the tilting of the ectodomain to make the MPER sterically accessible to the Fab and (2) MPER motions that sufficiently expose the epitope from the membrane to favor Fab binding. Our use of the structurally well characterized bnAb 4E10 that recognizes the MPER hinge and C-helix[25,26] made it possible to test this notion. Incubation of the nanodisc-embedded gp145 with a 100-fold molar excess of the 4E10 Fab resulted in ~20% of the vitrified

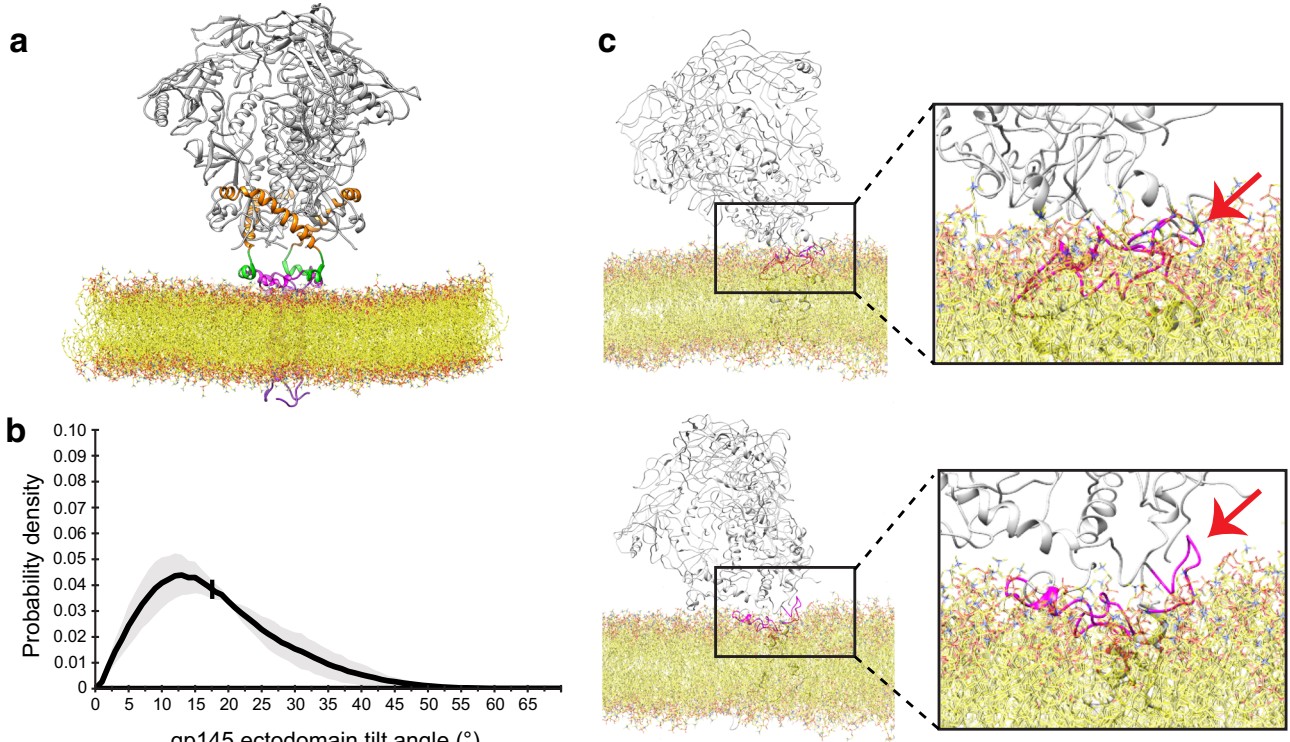

**Fig. 2 | Molecular-dynamics simulations of nanodisc-embedded HIV-1 gp145.**
**a** The system of membrane-embedded gp145 used for coarse-grained molecular-dynamics (CGMD) simulations. The gp145 ectodomain is shown in gray, CHR in orange, MPER-N in green, and MPER-C in magenta. The lipid bilayer is shown in yellow. **b** Graph showing the distribution of tilt angles adopted by the gp145 ectodomain, summarizing data from all five repeats. The solid line represents the mean and the gray band indicates the standard deviation. The vertical line at 18° is the overall average tilt of the ectodomain. **c** Two snapshots from the CGMD simulations converted into all-atom views showing two highly tilted gp145 ectodomains. The tilt of the ectodomain does not appear to correlate with the opposite MPER-C segment (arrow) being either lipid-immersed (top panel) or exposed (bottom panel).

ectodomains being bound by at least one Fab molecule. Projection averages obtained by 2D classification of negatively stained particles revealed that one to three Fabs could be bound to a gp145 (Fig. 3a). After extensive processing of the vitrified Fab-bound particles, we were able to generate density maps of nanodisc-embedded gp145 in complex with one 4E10 Fab (gp145•1Fab) at 8.8-Å resolution, with two 4E10 Fabs (gp145•1Fab) at 8.2-Å resolution and with three 4E10 Fabs (gp145•1Fab) at 3.7-Å resolution (Fig. 3b and Supplementary Fig. 2). Despite the limited resolution of two of the maps, which is only ~12 Å in the Fab-binding region (Supplementary Fig. 4b), the features of the additional density revealed these to represent bound Fabs and allowed for docking of the 4E10 Fab crystal structure (PDB: 1TZG)[25] (Supplementary Fig. 6). Remarkably, the maps reveal two distinct modes in which the MPER-specific 4E10 Fab binds to gp145. The gp145•1Fab map illustrates binding mode 1, in which the Fab extends from the nanodisc almost parallel to the membrane surface but slightly away from the membrane plane in the same direction as the ectodomain and with the plane of the Fab parallel to that of the membrane. In the gp145•2Fab map, one 4E10 Fab also exhibits binding mode 1, but the second Fab is bound in mode 2, in which the Fab extends in the direction opposite that of the ectodomain, occupying space that would normally be that of the lipid bilayer. In addition, the plane of the Fab is rotated by ~90° and is thus almost perpendicular to the membrane plane. Finally, in the gp145•3Fab map, all three Fabs are bound to gp145 in binding mode 2. As all three Fabs now occupy the space that would normally be occupied by the membrane, this map shows no density that would represent the nanodisc.

To analyze the medium-resolution maps of nanodisc-embedded gp145 in complex with one and two 4E10 Fabs, we docked the 4E10 Fab in complex with the MPER-C peptide (PDB: 1TZG)[25] (Supplementary Fig. 6). The long complementarity-determining region 3 in the heavy chain (CDRH3) of 4E10 is highly hydrophobic, and 4E10 utilizes its CDRH1 and CDRH3 loops to bind lipids[26], implying that those CDRs form an interaction surface with the lipid bilayer. In our structure of the nanodisc-embedded gp145 by itself, the ectodomain is oriented perpendicular to the membrane plane (Fig. 4a). However, for the 4E10 Fab to be able to bind to gp145, the ectodomain must be tilted by ~25° (Fig. 4b). A similar tilt angle has recently been seen in a cryo-EM structure of the full-length Env protein of the AMC011 strain in complex with the Fab of VRC42.01[21]. The ectodomain in our gp145•2Fab structure showed a similar tilt angle of ~25° (Fig. 4c). We also measured the distance of the gp145 ectodomain from the membrane surface. For gp145 by itself, the distance between Glu662 (the end of the C-terminal heptad repeat; CHR) and Trp666 (the beginning of the membrane-immersed MPER) is ~10 Å. In the gp145•1Fab structure, this distance increases to ~20 Å. In the gp145•2Fab structure, the distance increases further to ~40 Å for the second Fab. We found a similar distance of ~40 Å when we docked the atomic models of Env and the complex of the 10E8 Fab bound to its MPER epitope (PDB: 4U6G) into the published cryo-EM map of the Env ectodomain from the AMC011 strain with two bound 10E8 Fabs (Fig. 4d)[21]. Collectively, these findings show that the Env ectodomain in complex with MPER-targeting bnAbs is tilted and lifted away from the membrane.

The MPER-C segment (residues 674–683) bound by the 4E10 Fab adopts a substantially different orientation than in the unbound state. In our structure of nanodisc-embedded gp145, the MPER-C segments should run mostly parallel to the membrane and be partially immersed in the lipid bilayer (Fig. 4a). In contrast, our model for the gp145•1Fab complex indicates that the MPER-C segment bound by the first Fab in binding mode 1 is extracted from the membrane, with the C-terminal end of the MPER remaining close to the membrane surface but the MPER-C segment now being oriented at an ~80° angle with respect to

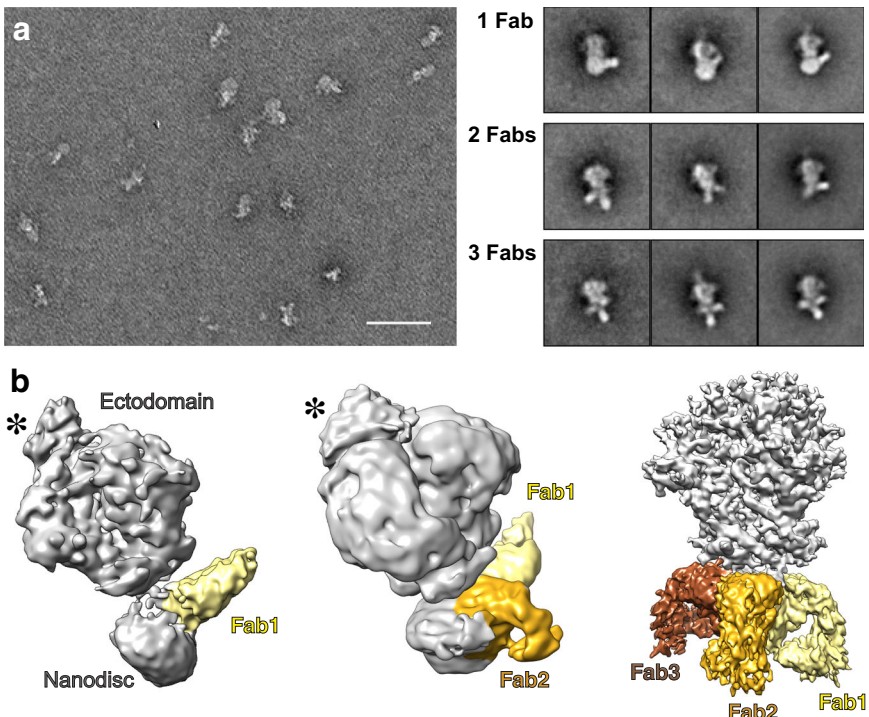

**Fig. 3 | Cryo-EM analysis of nanodisc-embedded gp145 in complex with the Fab of anti-MPER bnAb 4E10. a** Representative negative-stain EM image of nanodisc-embedded gp145 after incubation with 4E10 Fab (left) and 2D-class averages showing gp145 with one to three bound 4E10 Fabs (right). Scale bar: 50 nm; side length of averages: 34 nm. **b** Cryo-EM maps of gp145 with one, two, and three bound 4E10 Fabs. The maps were automatically segmented with the "color zone" command in Chimera. The ectodomain and nanodisc are colored in light gray and Fabs 1 to 3 are colored in yellow, gold, and brown, respectively. Asterisks indicate density representing the PG9 Fab used for purification.

the membrane plane (Fig. 4b). Binding of the second Fab in binding mode 2 results in a further removal of the MPER-C segment that now runs almost parallel to the membrane but at a distance close to 40 Å (Fig. 4c). This position of the MPER-C segment would require extraction of the TM helix and is therefore unlikely to occur in situ, but it does imply that Fab binding exerts a substantial pulling force on the TM helix, likely resulting in a disturbance of the transmembrane region and possibly even some extraction of the helix out of the membrane. Of note, the drastic reorientation of the MPER-C segment seen in the binding of the second Fab in binding mode 2 is not observed in the modeled structure of the Env ectodomain with two bound 10E8 Fabs (Fig. 4d). In this structure, both MPER-C segments adopt a conformation like that observed for the 4E10 Fabs bound in binding mode 1 in the gp145•1Fab and gp145•2Fab structures. The difference in 4E10- and 10E8-bound structures may be related to the greater hydrophobicity and lipid binding of the membrane-interaction surface of 4E10[14,27] and/or to the truncated C-terminal domain of gp145 used for complex formation with 4E10.

The 3.7-Å resolution of the gp145•3Fab map enabled us to build an atomic model for the ectodomain as well as the entire MPER for one of the subunits and backbone models for the MPERs of the other two subunits, thus revealing how the MPER is connected to the former (Fig. 4e, Supplementary Figs. 4f and 7a). However, we observed neither density for the nanodisc or the TM helices, suggesting that binding of three 4E10 Fabs results in the complete extraction of the trimer from the nanodisc. Prior direct measurement of 4E10 Fab binding to MPER peptide on HIV-1 virus mimetic liposomes by isothermal titration calorimetry revealed an enthalpy change of −25 kcal/mol[14]. That exothermic process in association with a weak binding constant of 1.0 μM $K_d$ as determined by surface plasmon resonance suggests a significant entropic penalty. While we currently have no direct energetic measurements for the extraction of the gp41 TM by 4E10 in the nanodisc context, single-molecule atomic-force microscopy methods reveal

that even multi-pass transmembrane proteins can be unfolded and extracted from the membrane at forces (pN) on par with or below those mediated by adhesion molecules[28]. Even though our gp145•3Fab map (further described in Supplementary Material) clearly represents an artificial situation, it allows us to draw two conclusions. First, in the context of the ectodomain, the MPER epitope recognized by the 4E10 bnAb remains in the same conformation as observed in the crystal structure of the 4E10 Fab in complex with the MPER peptide (Supplementary Fig. 7b)[26]. Second, the complete membrane extraction of the Env trimer upon binding of three 4E10 Fabs illustrates the tension 4E10 binding exerts on the transmembrane domain, which may explain the distorted appearance of the nanodisc in our gp145•1Fab map (Fig. 3b).

### Effect of bnAb binding on Env dynamics

To understand the effect of bnAb binding on Env dynamics, we analyzed the CGMD simulations of the membrane-embedded gp145 and converted snapshots of highly tilted ectodomains to atomistic models. In many of the models, the conformation of the MPER-C segment was too different from that seen in the crystal structure of the 4E10 Fab complex to allow unambiguous docking (Supplementary Fig. 8a). This finding suggests that the 4E10 epitope may only occasionally be in an optimal conformation for antibody binding. Alternatively, or in addition, it is conceivable that initial interactions of the antibody can be with only part of its epitope as suggested previously[13] and that this then catalyzes folding of the entire MPER-C segment to allow for full engagement of the Fab with its entire epitope as seen in the crystal structure[25,26]. Furthermore, in almost all the models, the MPER-C segment was oriented in a way that docking of the 4E10 Fab either resulted in the Fab being partially embedded in the membrane or in sterical clashes with the ectodomain (Supplementary Fig. 8b). However, one snapshot showed the MPER-C segment in a similar orientation as seen in our gp145•1Fab structure and its conformation allowed for docking

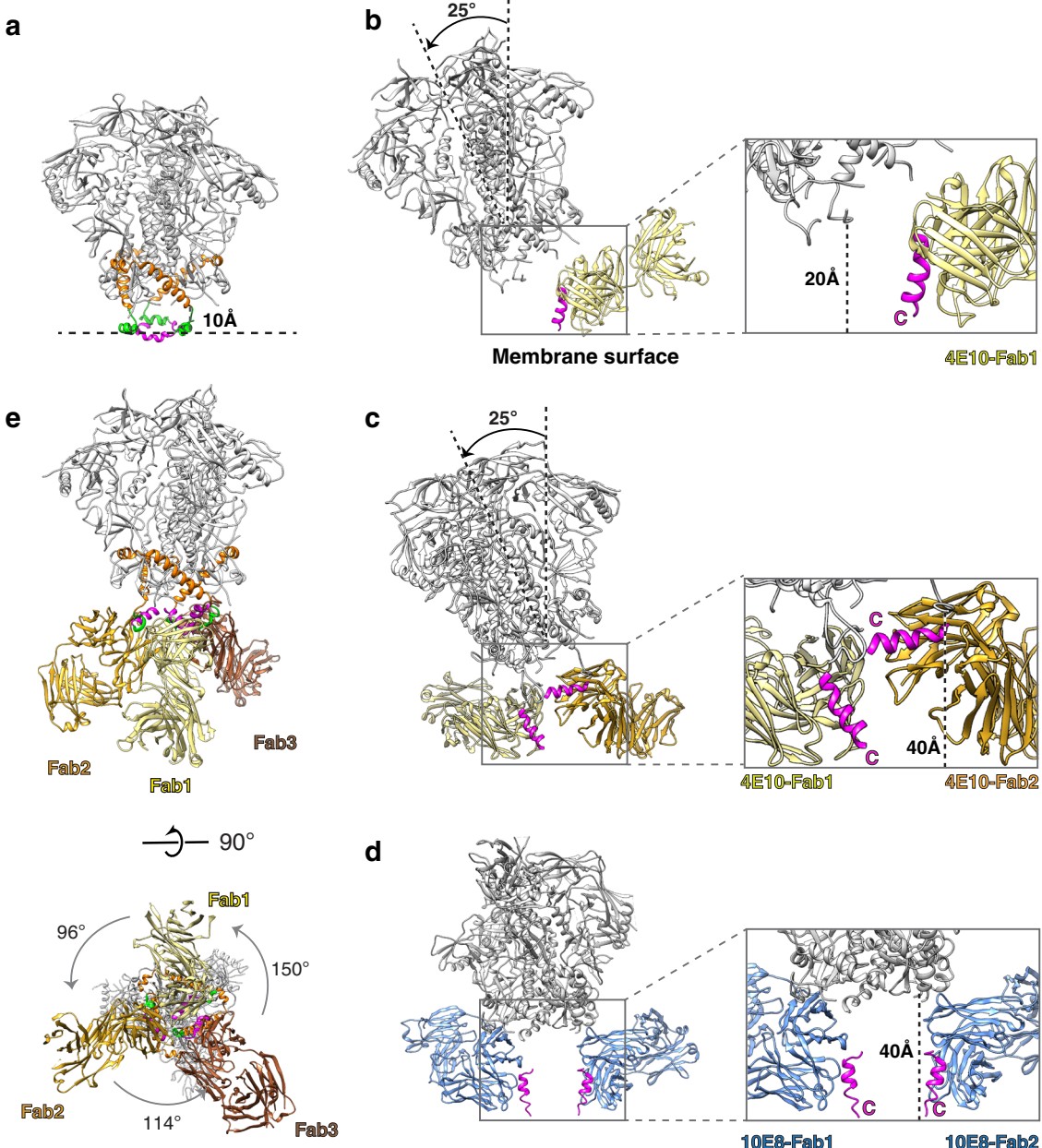

**Fig. 4 | Conformational changes in gp145 upon binding of 4E10 Fabs. a** The cryo-EM model of gp145 by itself (ectodomain in gray, CHR in orange, and MPER-N in green) with added MPER-C segments from an NMR structure (PDB: 2PV6) showing that the MPERs are membrane-embedded and the distance of the ectodomain (residue Ala662) from the membrane surface is ~10 Å. **b** Model of gp145 with one 4E10 Fab bound (colored in yellow), showing that the ectodomain is tilted, increasing its distance from the membrane to ~20 Å, and that the MPER-C segment (magenta) must be exposed and oriented approximately perpendicular to the membrane plane (binding mode 1). **c** Model of gp145 with two 4E10 Fab bound (colored in yellow and gold), showing that the ectodomain remains tilted and that the two Fabs bind very differently. Whereas one Fab shows the same binding mode 1 as in panel (**b**), the second Fab lifts the MPER-C segment to ~40 Å from the membrane, where it runs approximately parallel to the membrane (binding mode 2); this Fab occupies space normally occupied by the membrane. **d** Model of gp160 with two bound Fabs of bnAb 10E8 (colored in blue) generated by docking the crystal structure of the 10E8 Fab–MPER epitope complex (PDB: 4U6G) into the cryo-EM map of 10E8 Fab bound to the Env protein of the AMC011 strain (EMDB: 21334). The model shows that the ectodomain is not tilted and yet ~40 Å away from the membrane surface and that both 10E8 Fabs bind MPER-C similar to 4E10 binding mode 1. **e** Cryo-EM structure of gp145 with three 4E10 Fabs bound (colored in yellow, gold and brown) seen parallel (top) and perpendicular to the membrane plane (bottom). Although binding asymmetrically, all three Fabs show binding mode 2. The color coding is the same as in Figs. 1–3.

of the 4E10 Fab (Fig. 5a). CGMD simulations of the membrane-embedded gp145•1Fab complex showed the ectodomain to assume a similarly wide range of angles as seen in the absence of the 4E10 Fab (Fig. 5b and Supplementary Movie 4), but with an average tilt angle of 28.7° ± 12.0°, different from that of the unliganded ectodomain (17.6° ± 10.0°). Furthermore, while the three MPER-C segments in the unliganded gp145 have an average tilt of 29.0° ± 20.7° (Supplementary Fig. 9) but appear to move independently of each other (Fig. 5c and

Supplementary Fig. 10), in the case of 10E8 Fab-bound gp145, the MPER-C segments have more distinct angles (Fig. 5d and Supplementary Fig. 11). The Fab-bound MPER-C is stabilized at a high tilt of 64.9° ± 10.4°, similar to the one derived from the cryo-EM map, ~80° (Fig. 4b), whereas the neighboring MPER adopts an intermediate tilt of 42.5° ± 15.3°, potentially enough to facilitate binding of a second Fab, and the last MPER assumes a low tilt angle of 20.6° ± 12.7°, similar to that of the MPER-C segments in the unliganded gp145.

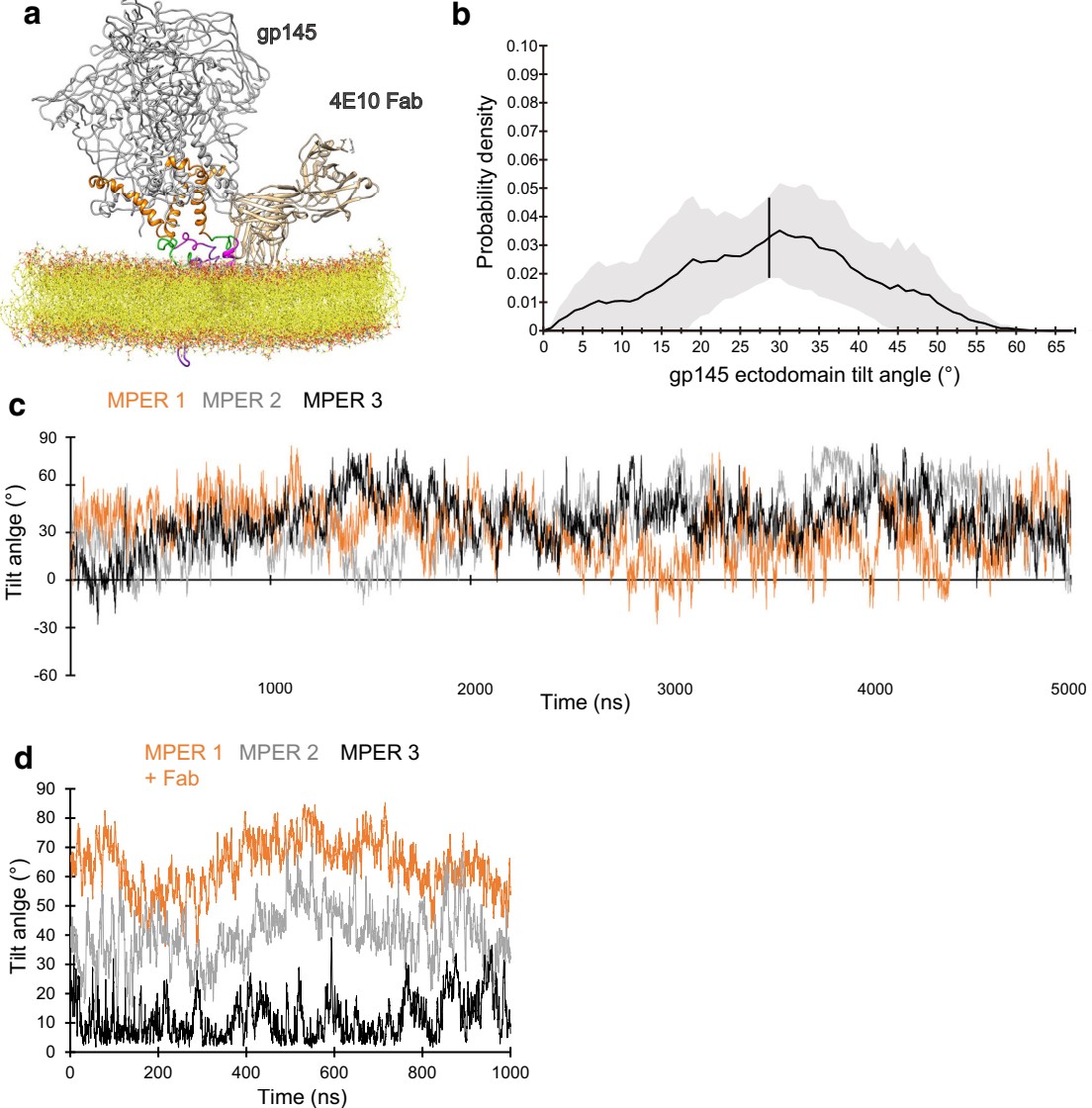

**Fig. 5 | Molecular-dynamics simulation of membrane-embedded gp145 with bound 4E10 Fab. a** The system of membrane-embedded gp145 with bound 4E10 Fab used for coarse-grained molecular-dynamics (CGMD) simulations. The gp145 ectodomain is shown in gray, CHR in orange, MPER-N in green, and MPER-C in magenta. The bound Fab is shown in gold and the lipid bilayer in yellow. **b** Graph showing the distribution of tilt angles adopted by 4E10 Fab-bound gp145 ectodomain, summarizing data from all five repeats. The solid line represents the mean, and the gray band indicates the standard deviation. The vertical line at 29° is the overall average tilt of the ectodomain. **c** Angles of the three MPER-C segments relative to the membrane plane adopted over time in one of the CGMD simulations of unliganded gp145, illustrating the highly dynamic and uncorrelated movements of the three segments (see Supplementary Fig. 10 for data from all repeats). **d** Angles of the three MPER-C segments adopted over time in one of the CGMD simulations of 4E10 Fab-bound gp145, illustrating that Fab binding stabilizes the three MPER-C segments at different average angles (see Supplementary Fig. 11 for data from all repeats).

## A mechanistic basis for potent anti-MPER bnAb-mediated inhibition of viral fusion

Our structures of nanodisc-embedded gp145 by itself and in complex with up to three 4E10 Fabs shed light on the prerequisites and consequences of bnAb binding to the MPER (Fig. 6a). For most of the time, the highly glycosylated Env ectodomain will shield the MPER whose bnAb epitope is immersed in the membrane to a large extent. However, our cryo-EM and CGMD analyses reveal that the ectodomain can adopt a wide range of angles with respect to the membrane. At a tilt angle of ~25° (the angle seen in the cryo-EM map of 4E10 Fab-bound gp145) or higher, which occurs for ~24% of the time (deduced from the CMGD simulations of unliganded gp145), an MPER becomes accessible to Fab binding, which also and independently requires the epitope to have sufficient exposure on the membrane and adopt a conformation that allows for at least initial binding interactions with the antibody. After engagement of the antibody with the partially exposed MPER, complete binding will stabilize the MPER in a membrane-extracted configuration thereby disrupting the independent and comparable motions of the MPER protomers that otherwise facilitate downstream hemifusion and fusion processes (see below). Overlay of the epitope-bound Fabs of other MPER-targeting bnAbs with the 4E10 Fab guided by their respective epitopes suggests that this may be a general principle for the neutralizing activity of these bnAbs (Fig. 6b). Antibody binding also interferes with Env function by exerting tension on the transmembrane domains. While our maps do not reveal the effect this tension has on the TM domains, a previous study proposed that antibody binding might disrupt the putative helical bundle formed by the TM domains[21]. Furthermore, the capacity of the anti-MPER bnAbs to

bind unliganded as well as receptor-bound Env configurations and the pre-hairpin intermediate state (Supplementary Fig. 12) provides an extended time window to mediate viral neutralization.

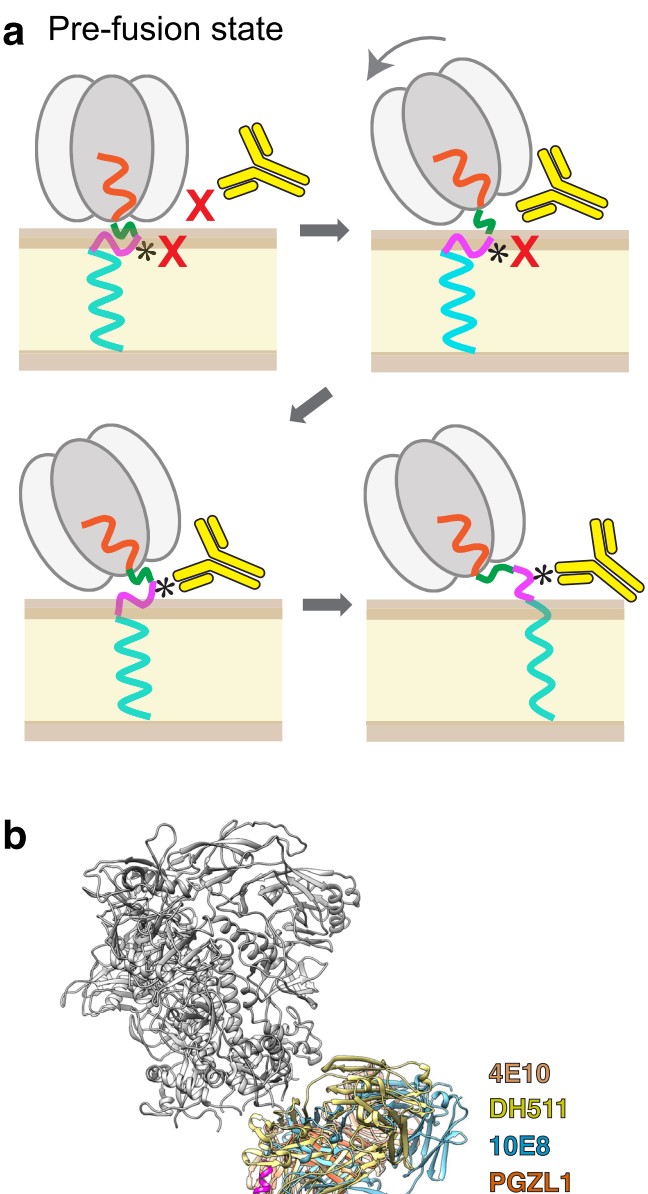

**Fig. 6 | Model for stepwise binding of an anti-MPER bnAb to the HIV-1 Env protein.** The gp145 ectodomain is shown in gray, CHR in orange, MPER-N in green, MPER-C in magenta, and the transmembrane domain in light blue. The lipid bilayer is shown in pale yellow/brown and the antibody in bright yellow. **a** In the upright orientation of the Env protein, the MPER epitope for bnAb 4E10 (indicated by *) is occluded by the ectodomain, making it inaccessible to the bnAb, and is most of the time not exposed but buried in the membrane. Tilting of the ectodomain transiently makes the opposing MPER accessible and if this coincides with at least partial exposure of the epitope, the bnAb can make first interactions. Full antibody binding both alters MPER motion on the membrane and creates strain on the transmembrane domains, thus preventing further conformational changes required for membrane fusion. **b** Structural alignment of Fabs of MPER-targeting bnAbs bound to HIV-1 Env. The Fabs of bnAbs 4E10 (PDB: 4XC3), DH511 (PDB: 5U3N), 10E8 (PDB: 4U6G), and PGZL1 (PDB: 6O3J) were aligned based on the consensus MPER epitope (magenta), showing that all MPER-C targeting bnAbs bind to HIV-1 Env in a similar fashion. Note that the lipid-interaction surface of 4E10 is more hydrophobic than that of the other bnAbs[26].

## Discussion

The tilting and twisting motions of the HIV-1 trimer ectodomain observed experimentally are consistent with MD simulations. While motions can be anticipated for ectodomains of membrane receptors, particularly those with single-pass TM segments, we are unaware of comparable empirical findings reported in other systems. A previous cryo-EM study of Env protein in membrane-mimetic environments also observed tilting of Fab-bound Env, but interpreted the observed tilting to be mostly the result of Fab binding[21]. In contrast, a recent cryo-ET study of native HIV-1 virions published after completion of this work also found that Env proteins adopt different tilt angles relative to the virus membrane[29], thus independently corroborating our single-particle cryo-EM and CGMD results that the Env protein undergoes tilting motions in the absence of bound antibody. In this regard, it is conceivable that coordination of the orthogonal docking to gp120 by the HIV-1 receptor CD4 and its co-receptor (CXCR4 or CCR5)[30] that prime and initiate conformational change required for gp160-mediated fusion[31], respectively, pose challenges mitigated by ectodomain gesticulation.

That said, it is worth emphasizing that during MD simulations the trimer resides close to the viral membrane, occluding the tripod, for the majority of the analysis. Fewer than 10% of amassed single particles were used for the 3.9-Å resolution map of the nanodisc-embedded trimer without 4E10 Fabs bound. Unsurprisingly, the others were too heterogeneous to permit visualization of the MPER tripod, a fact underscoring the notion that the MPER is flexible on the membrane. This view of the MPER as being largely membrane-immersed is also consistent with the proximity of the gp160 ectodomain to the membrane forming a 10-Å crawlspace noted herein (Fig. 1) and in independent cryo-ET data[29]. These two findings contrast with those deduced from another cryo-ET study of HIV-1 BaL Env trimers on aldrithiol-2-inactivated viral particles, in which the MPER appears to form a stalk-like link between the Env ectodomain and the viral membrane[32]. The basis for this disparity is currently unknown.

Prior NMR studies across viral clades identified a structurally conserved pair of viral lipid-immersed N- and C-helices separated by a hinge with tandem joints locked by requisite helix capping residues identified from sequence analysis of more than 20,000 viral HIV-1 strains. Disruption of both joints by alanine substitution of those caps abrogated both hemifusion and fusion mediated by CD4-dependent as well as CD4-independent strains[13]. By concomitantly covering the viral membrane with these mobile hydrophobic MPER segments, less dehydration of the viral membrane is required to physically approximate solvated lipid bilayers to mediate fusion, thereby reducing the requisite high kinetic barrier[31,33]. In addition, the MPER-TM segment per se induces membrane curvature and lipid mobility also involved in membrane hemifusion and fusion[33]. These fusion facilitative features of the MPER are noteworthy, given the order of magnitude lower number of spikes per HIV-1 virion compared to multiple other viruses including influenza A[34]. Because binding of a single Fab impacts the membrane surface motions not only of the bound MPER but also of those of the other two protomers (Fig. 5d and Supplementary Fig. 11) and with binding of one antibody or its Fab fragment appearing sufficient to inactivate the trimer[35], this alteration of the MPER likely contributes importantly to neutralization.

Considering the vital role of the MPER at all stages of HIV-1 fusion, the virus' rationale to shield its tripod structure from immune antibody attack is evident. While spike tilting, and possibly CD4 priming, grants transient access to an antibody Fab arm, the immunodominance of other viral ectodomain sites is an effective diversionary strategy to mitigate MPER targeting[35,36]. Our prior MPER immunogenicity studies revealed that while a membrane surface-arrayed MPER was immunogenic in vivo[37], the approach angle of the majority of vaccine-elicited antibodies is incompatible with the steric block imposed by the ectodomain. A future vaccine strategy is to restrict the antibody approach

angles to those adopted by existing bnAbs (Fig. 6b) to bind the native MPER during tilting. The enormous difference between vaccine effectiveness against two RNA viruses, namely HIV-1 *versus* CoV-2, are associated with high rate of mutations and the density of glycan shielding of the former *versus* the latter[38]. Targeting the most highly conserved and accessible site of HIV-1 therefore seems critical for the development of protective vaccines against HIV-1, given the rationale defined herein.

## Methods

### Antibody expression and purification

The PG9 HRV3C IgG was expressed in Freestyle™ 293-F cells (Thermo Fisher Scientific) cultured in suspension using FreeStyle™ 293 expression medium. A 1-L culture of Freestyle™ 293-F cells was transfected with a 2:1 (w/w) mixture of HC: LC PG9 DNA using PEI 'Max' (Polysciences, Inc). After 6 days, cells were pelleted, and the supernatant was loaded onto a Protein A-Sepharose column. The IgG was eluted using 0.5 M acetic acid, the solution neutralized with 3 M Tris-HCl, pH 9.0, and dialyzed overnight against phosphate-buffered saline (PBS; 8 mM $Na_2HPO_4$, and 2 mM $KH_2PO_4$, pH 7.4, 137 mM NaCl, 2.7 mM KCl).

The 4E10 IgG was expressed in Expi™ 293-F cells (Thermo Fisher Scientific) cultured in suspension using Expi™ 293 expression medium. A 1-L culture of Expi™ 293-F cells was transfected with a 1:1 (w/w) mixture of HC: LC 4E10 DNA using PEI 'Max' (Polysciences, Inc). After 4 days, cells were pelleted, and the supernatant was loaded onto a Protein G-Sepharose column. The IgG was eluted using 500 mM acetic acid, the solution neutralized with 3 M Tris-HCl, pH 9.0, and dialyzed overnight against PBS. The 4E10 Fab was prepared by reducing 4E10 IgG (15 mg/mL) in 100 mM dithiothreitol for 1 h at 37 °C, followed by alkylation in 2 mM iodoacetamide for 48 h at 4 °C. The IgG was then digested with endoproteinase Lys-C (0.01 µg/µL; Roche Applied Sciences) in 25 mM Tris-HCl, pH 8.5, and 1 mM EDTA for 4 h at 37 °C. The cleavage reaction was stopped with 1 mM Nα-*p*-tosyl-l-lysine chloromethyl ketone (TLCK) and 0.4 mM leupeptin, and the cleavage products were passed over a Protein A-Sepharose column to remove Fc and intact IgG.

### gp145 protein expression

The DNA sequence encoding residues 1–723 of the Env protein from HIV-1 strain BG505, containing three stabilizing mutations (Ala501Cys, Thr605Cys, Ile559Pro), an introduced glycosylation site (Thr332Asn) and an improved furin cleavage site (REKR to RRRRRR), and the furin gene were individually cloned into pEG BacMam expression vectors. The plasmids were transformed into *Escherichia coli* DH10Bac cells to generate bacmid DNA. Recombinant baculovirus was produced through three rounds of viral amplification in *Spodoptera frugiperda* Sf9 cells cultured in Sf-900III SFM medium at 27 °C. For protein expression, the baculoviruses containing gp145 and furin DNA, respectively, were mixed at a ratio of 2:1 (v/v) and used to infect Freestyle™ 293-F cells at a ratio of 1:10 (v/v). After 24 h of infection, sodium butyrate was added to the cultures at a final concentration of 10 mM, the cultures were moved from 37 °C to 30 °C and allowed to grow for an additional 48 h. Cells were harvested and washed in PBS supplemented with 5 mM EDTA and 0.1% (w/v) bovine serum albumin (Sigma Aldrich).

### gp145 purification

The gp145 trimer was purified based on published protocols[39] with slight modifications. Freestyle™ 293-F cells expressing gp145 were incubated with PG9 IgG for at least 3 h, washed with PBS, and solubilized with lysis buffer containing 50 mM Tris-HCl, pH 7.5, 150 mM NaCl, 0.5% Triton X-100 and protease inhibitor cocktail (Roche). After centrifugation of the cell lysate at 38,400 × *g* for 1 h at 4 °C, the supernatant was incubated overnight with Protein A resin (GenScript) at

4 °C. The resin was successively washed with 10 column volumes (CV) of wash buffer 1 (50 mM Tris-HCl, pH 7.5, 150 mM NaCl, 0.03 mg/mL sodium deoxycholate, 0.1% w/v CHAPS), 10 CV of wash buffer 2 (50 mM Tris-HCl, pH 7.5, 500 mM NaCl, 0.03 mg/mL sodium deoxycholate, 0.1% (w/v) n-dodecyl β-D-maltoside (DDM)), and 10 CV of wash buffer 3 (50 mM Tris-HCl, pH 7.5, 150 mM NaCl, 0.03 mg/mL sodium deoxycholate, 0.1% DDM, 2 mM EDTA). The PG9 IgG bound to the Protein A resin was digested into Fabs with 3 C protease in wash buffer 3 supplemented with 80 mM L-cysteine (Sigma Aldrich) for 4 h at 4 °C. The eluate was collected, together with a 5 CV wash with SEC buffer (50 mM Tris-HCl, pH 7.4, 150 mM NaCl, 0.03 mg/mL sodium deoxycholate, 0.05% DDM). The protein solution was concentrated using Amicon Ultra 15-mL 100-kDa cut-off centrifugal filters (Millipore Sigma) and run over a Superose 6 size-exclusion column (GE Healthcare) using SEC buffer.

### Reconstitution of gp145 into nanodiscs

The lipids used in this study are brain polar lipid extract, palmitoyl oleyl phosphatidylcholine (POPC) and palmitoyl oleyl phosphatidylglycerol (POPG), all purchased from Avanti Polar Lipids. The lipids were solubilized with 20 mM sodium cholate in 30 mM Tris-HCl, pH 7.5, 150 mM NaCl with sonication and mixed at a molar ratio of 1.07:1.5:1. For reconstitution into nanodiscs, gp145 purified in DDM, scaffold protein MSP1D1dH5 and the lipid solution were mixed at a molar ratio of 1:20:300. After a 30-min incubation on ice, Bio-Beads SM-2 (Bio-Rad) were added to 30% of the sample volume to remove the detergents. The Bio-Beads were replaced after 3 h. After the mixture was incubated overnight at 4 °C with constant rotation, the Bio-Beads were allowed to settle by gravity and the supernatant containing nanodisc-embedded gp145 was collected.

### Decoration of nanodisc-embedded gp145 with 4E10 Fab

Nanodisc-embedded gp145 was incubated with 4E10 Fab at a molar ratio of 1:100 for at least 3 h. Samples were concentrated using Amicon Ultra 15-mL 50-kDa cut-off centrifugal filters (Millipore Sigma) and loaded onto a Superose 6 size-exclusion column equilibrated with 50 mM HEPES, pH 7.4, and 150 mM NaCl. Peak fractions containing nanodisc-embedded gp145 decorated with 4E10 Fab were pooled.

### Negative-stain EM analysis

The homogeneity of samples was first assessed by negative-stain EM using 0.7% (w/v) uranyl formate as described[40]. To calculate negative-stain EM averages, 100 images were collected for each sample using an XR16L-ActiveVu charge-coupled device camera (AMT) on a Philips CM10 electron microscope (Philips) operated at an acceleration voltage of 100 kV. The calibrated magnification was ×41,513 (nominal magnification of ×52,000), yielding a pixel size of 2.65 Å at the specimen level. The defocus was set to −1.5 µm. Approximately 10,000 particles were manually selected for each sample using the e2boxer.py command of the EMAN2 software package[41] and windowed into 180 × 180-pixel images. After image normalization and particle centering, the particle images were classified into 100 groups using K-means classification procedures implemented in SPIDER[42].

### Cryo-EM sample preparation and data collection

The nanodisc-embedded gp145–4E10 Fab complex was cross-linked with BS3 at a molar ratio of 1:600 for 30 min on ice. Cross-linking was terminated by adding 50 mM Tris, pH 7.5. The sample was dialyzed against buffer with 50 mM Tris, pH 7.5, 150 mM NaCl to remove the cross-linker. Homogeneity of the sample was examined by negative-stain EM. For cryo-EM, the sample concentration was measured with a NanoDrop spectrophotometer (Thermo Fisher Scientific) and adjusted to 0.2 mg/mL. Aliquots of 4 µL were applied to mildly glowed-discharged graphene oxide (GO) grids (GO on Quantifoil R1.2/1.3, Cu, 400 mesh, Electron Microscopy Sciences) using a Vitrobot Mark VI

(Thermo Fisher Scientific) set at 4 °C and 100% humidity. After 20 s, grids were blotted for 1 s with a blot force of −2 and plunged into liquid nitrogen-cooled ethane. Cryo-EM datasets were collected on a 300-kV Titan Krios electron microscope (Thermo Fisher Scientific) equipped with a K2 Summit electron detector (Gatan) at a nominal magnification of ×29,000 in super-resolution counting mode. After binning over 2 × 2 pixels, the calibrated pixel size was 1.03 Å on the specimen level. Exposures of 10 s were dose-fractionated into 40 frames (0.25 s per frame) with a dose rate of 8 e⁻/pixel/s, resulting in a total dose of 80 e⁻/Å². Data were collected using SerialEM[43] in 'superfast mode', in which 3 × 3 holes are exposed using beam tilt and image shift before moving the stage to the next position[44]. The defocus range was set from −1.5 to −2.5 μm. Data collection parameters are summarized in Supplementary Table 1.

### Cryo-EM data processing
The 30,404 movie stacks from five data collection sessions were gain-normalized, motion-corrected, dose-weighted, and binned over 2 × 2 pixels in MotionCorr2[45]. The contrast transfer function (CTF) parameters were estimated with CTFFIND4 (ref. [46]) implemented in RELION-3[47]. Particles were automatically picked without templates with Gautomatch (http://www.mrc-lmb.cam.ac.uk/kzhang/Gautomatch/), extracted into individual images, normalized and subjected to 2D classification into 100 classes in RELION-3. Classes with good 2D averages were combined, yielding 3,039,896 particles.

Identification of particles with bound 4E10 Fabs was complicated by the dominance of the density of the gp145 ectodomain. Therefore, only particles from 2D classes showing clear side views were selected and used to generate four initial models using the ab initio algorithm in cryoSPARC v2[48]. The ab initio reconstruction generated one map showing clear density of an ectodomain with three bound 4E10 Fabs, one map showing strong density only for the ectodomain, and two "junk" maps. The first two maps and one of the junk maps were used as initial models to run two rounds of heterogeneous refinement in cryoSPARC v2, resulting in a stack of 362,646 particles showing clear density for three bound 4E10 Fabs, and 1,878,941 particles showing clear density only for the ectodomain.

The 362,646 particles showing clear density for three bound 4E10 Fabs were subjected to subsequent homogeneous and non-uniform refinements in cryoSPARC v2[49], yielding a final map at a global resolution of 3.66 Å (Supplementary Fig. 4a).

The 1,878,941 particles showing clear density only for the ectodomain were first subjected to a supervised classification in RELION-3, using two reference maps generated with the Segger tool in Chimera from the cryo-EM map of the nanodisc-embedded full-length HIV-1 Env protein from strain AMC011 with bound PGT151 Fab and 10E8 Fab (EMDB 21332)[21]. One reference map contained only density for the ectodomain and nanodisc, whereas the second reference map also included density for the 10E8 Fab. The 224,730 particles identified by this classification to have Fab bound were subjected to further 3D classification. The density for the PG9 Fab that was used for purification was always masked out. After a first 3D classification into eight classes, four classes showing clear Fab density were combined and subjected to a second round of 3D classification into four classes. Two of the resulting classes showed an ectodomain with one 4E10 Fab bound. These two classes were combined and subjected to two further rounds of 3D classification into four classes. The final map containing 23,538 particles was sharpened by post-processing to yield a density map at a global resolution of 8.8 Å (gp145•1Fab) (Supplementary Fig. 4a). One class of the initial 3D classification into four classes showed an ectodomain with two 4E10 Fabs bound. This class contained 21,454 particles and the map was sharpened by post-processing to yield a map at a global resolution of 8.2 Å (gp145•2Fab) (Supplementary Fig. 4a).

The remaining 1,654,211 particles from the initial supervised classification showing no density for 4E10 Fab were subjected to further supervised classification using as references two maps of Env with and without nanodisc density. The 1,153,408 particles showing density for the nanodisc were subjected to five rounds of 3D classification, at each step of classification, the classes with clear density for both ectodomain and nanodisc were selected and subjected to further classification. Finally, 47,616 particles from two classes showing the best density in the MPER region were combined and refined, followed by CTF refinement and Bayesian polishing[50]. The resulting density map was sharpened by post-processing to obtain a final map at a global resolution of 3.9 Å (Supplementary Fig. 4a).

### Movie generation
The movie illustrating the range of orientations the ectodomain can assume on the nanodisc surface was generated from experimental maps. The maps were selected from the successive rounds of 3D classification of the 1,654,211 particles showing no density for 4E10 Fab. The maps were selected based on having a particle number of 1000 particles or less (to identify as many different orientations as possible) as well as on having a well-defined nanodisc density (to be able to determine the orientation of the ectodomain with respect to the nanodisc) (Supplementary Fig. 3). These criteria yielded 20 maps that were used to generate the movie in Chimera[51].

### Model building and refinement
For gp145 alone, the crystal structure of the Env ectodomain (residues 31–662, PDB: 5I8H)[52] and the NMR structure of the MPER peptide N-segment (residues 663–671, PDB: 2PV6)[14] were manually docked into the gp145 density map using the "fit in map" command in Chimera. After merging the chains, the model was manually adjusted in Coot[53] and refined against the density using phenix.real_space_refine[54] with geometric and Ramachandran restraints maintained throughout. After removing the side chains in poorly defined density regions, the final model contains backbone and side-chain information for residues 331–662 and only the backbone fold for residues 663–671.

For the gp145•3Fab complex, the crystal structures of the Env ectodomain (residues 31–662, PDB: 5I8H)[52], the NMR structure of the MPER peptide N-segment (residues 663–671, PDB: 2PV6)[14] and of the 4E10 Fab in complex with the MPER-C peptide (residues 672–684, PDB: 4XC3)[26] were manually docked into the gp145•3Fab density map using the "fit in map" command in Chimera. After merging the chains, the model was manually adjusted in Coot and refined against the density using phenix.real_space_refine with geometric and Ramachandran restraints maintained throughout. After removing the side chains in poorly defined density regions, the final model contains backbone and side-chain information for residues 31–662, and only the backbone fold for residues 663–684 (except for keeping the side chains for residue 678 in chain B and residues 664, 671, and 674 in chain D.

The models were refined against half map 1, and FSC curves were then calculated between the refined model and half map 1 (work), half map 2 (free), and the combined map (Supplementary Fig. 4c, e).

### Coarse-grained molecular-dynamics simulations
For simulations of unliganded gp145, a model was built by linking the cryo-EM structure of the ectodomain (this study) to the NMR structure of the MPER-transmembrane domain including the first seven residues of the cytoplasmic domain (PDB: 6DLN). The protein was placed into a lipid bilayer consisting of POPC and cholesterol at molar ratio of 4:1, and the system was solvated in 150 mM NaCl using the insane.py script[55].

Coarse-grained simulations were set up using Martinize2 (https://github.com/marrink-lab/vermouth-martinize) and the latest MARTINI3 force field model[56]. Quaternary and tertiary structure was

retained by applying an elastic network to the CG model, with a force constant of 500 kJ mol$^{-1}$ nm$^{-2}$, and upper and lower cut-off distances of 0.9 nm and 0.5 nm, respectively[57]. All simulations were performed with GROMACS 2020.6 (refs. [58], [59]) and the leap-frog algorithm[60] was used to integrate Newtonian equations of motion with a 20 fs timestep. Simulations were performed using the Verlet cutoff-scheme for Neighbor Searching, updated every 20 steps[61]. Electrostatic interactions were calculated using Reaction-Field, with a Coulomb cut-off distance of 1.1 nm[62]. The velocity rescale thermostat was used to retain the temperature at 310 K[63]. Pressure coupling for the equilibration and production simulations was performed using the Berendsen[64] and the Parrinello–Rahman barostat[65], respectively. Five simulations, starting from unique random seeds, were performed with 4.75 ns equilibration phases and 10 μs production runs each.

For simulations of gp145 in complex with 4E10 Fab, a snapshot was taken from a simulation of unliganded gp145 that showed an MPER-C conformation similar to that in the cryo-EM model of the 4E10 Fab-bound gp145. This coarse-grained snapshot was converted into an all-atoms view using the backward.py script[66] and the CHARMM-GUI All-atom converter[67]. The resulting atomistic model was used to dock the structure of the 4E10 Fab (PDB: 4XC3), guided by the Fab-bound MPER-C peptide, with minor adjustments to avoid steric clashes with the gp145 ectodomain. The protein was placed into a lipid bilayer consisting of POPC and cholesterol at molar ratio of 4:1, and the system was solvated in 150 mM NaCl using the insane.py script.

Coarse-grained simulations were set up and run as described for unliganded gp145. An elastic network was applied with the same parameters as described above. Elastic network interactions were also applied between the 4E10 Fab and the MPER-C segment to maintain binding interactions. Five simulations, starting from unique random seeds, were performed with 6.75 ns equilibration phases and 1 μs production runs each.

Simulation trajectories were centered using *gmx trjconv. gmx gangle* was used to measure the angles of the gp145 ectodomain and the MPER-C segments. Trajectories were visualized using VMD v.1.9.4a12 (ref. [68]) and graphs were made using Microsoft Excel.

### Reporting summary
Further information on research design is available in the Nature Research Reporting Summary linked to this article.

## Data availability
The data that support this study are available from the corresponding authors upon reasonable request. The cryo-EM maps have been deposited in the Electron Microscopy Data Bank (EMDB) under accession codes EMD-25022 (gp145), EMD-25024 (gp145•1Fab), EMD-25025 (gp145•2Fab), and EMD-25045 (gp145•3Fab). The atomic coordinates have been deposited in the Protein Data Bank (PDB) under accession codes 7SC5 (gp145) and PDB-7SD3 (gp145•3Fab). Previously reported PDB accession codes used are as follows: 1TZG (4E10 Fab/MPER); 2PV6 (MPER peptide N-segment, NMR); 4U6G (10E8 Fab/MPER); 4XC3 (4E10 Fab/MPER-C peptide); 5I8H (Env ectodomain); 5U3N (DH511 Fab/MPER); 6DLN (MPER tripod); 6E8W (MPER stalk–bubble); 6O3J (PGZL1 Fab/MPER). Previously reported EMDB accession codes used are as follows: EMDB-21332 (AMC011 with bound PGT151 Fab and 10E8 Fab) and EMDB-21334 (AMC011 with bound 10E8 Fab).

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

## Acknowledgements

We thank M. Ebrahim, J. Sotiris, and H. Ng at the Evelyn Gruss Lipper Cryo-EM Resource Center of The Rockefeller University for assistance

with cryo-EM data collection. We thank Dr. P. Cesar Telles de Souza for providing us a MARTINI3 beta topology file for cholesterol. This work was supported by National Institutes of Health grant P01 AI126901 (E.L.R., T.W.).

## Author contributions

E.L.R. and T.W. conceived the study. S.Y., G.H., M.K., E.L.R., and T.W. designed the experiments. S.Y., Y.W., and J.C. prepared antibody Fabs. S.Y. performed all biochemistry and structural biology experiments. G.H. performed all molecular dynamics experiments. J.H.W., M.K., E.L.R., and T.W. supervised experiments. All authors analyzed the results. S.Y., E.L.R., and T.W. wrote the manuscript.

## Competing interests

The authors declare no competing interests.
