## [Peer Review File · Nature Communications]

Dynamic HIV-1 spike motion creates vulnerability for membrane-bound tripod to antibody attackReviewers' Comments:

Reviewer #1:

Remarks to the Author:

Using cryo-EM studies of HIV envelope trimers embedded in lipid nano-discs, the authors provide novel and important observations about the molecular dynamics of the membrane proximal external regions (MPER) of the envelope and its accessibility to broadly neutralizing antibodies. There is a clear discussion of the relationship between this observed structures and previously published models. The findings are likely to be of high interest to the field. I have only two minor suggestions:

- 1) Could the authors please comment of the entropic and enthalpic costs of the membrane extraction of buried parts of the envelope sequence?
- 2) Could the authors please discuss the issue of steric hindrance in the binding of intact antibody molecules relative to Fab fragments, particularly in cases where more than a single antibody binds?

Reviewer #2:

None

Reviewer #3:

Remarks to the Author:

This is an important study in which the structure of nanodisc-embedded spike protein was solved by cryo-electron microscopy and investigated with molecular-dynamics simulations. Structures of spike proteins with bound 4E10 bnAb Fabs revealed that the antibody binds exposed MPER, thereby altering MPER dynamics, modifying average ectodomain tilt, and imposing strain on the viral membrane and the spike's transmembrane segments, resulting in the abrogation of membrane fusion and informing future vaccine development. The manuscript is well written and clearly presented. There are only a few minor complaints.

When describing the nanodisc-embedded gp145 at an overall resolution of 3.9 Å it is stated that this resolution was sufficient to build an atomic model for the entire ectodomain, which seems overstated. Even though the model is described well in Methods, the map quality including the side chain densities should also be addressed briefly in Results.

Figure 6. From the aligned Fabs it cannot be seen that 4E10 has the biggest lipid-interaction area.

In Methods: ... parameters were determined with CTFFIND4 (ref 45)...

Reviewer #4:

Remarks to the Author:

The manuscript was well-written, and the tilting-related epitope accessibility mechanism seems compelling. My primary assigned task was to review the molecular dynamics (MD) simulation component of the manuscript, so I will add a few suggestions to the authors below.

- 1) For the reader interested in the biophysics of HIV-1 Env, I'd suggest that the authors briefly mention the approximate lipid composition of the experimental nanodiscs and simulation membranes prior to the Methods section, it was the first question that popped into my mind as I was reading about the tilt angles.
- 2) I think it would be worth mentioning in the manuscript that the lipidome of the HIV-1 envelope has been published, and is quite a bit richer in cholesterol than the composition used for either the experiments or the simulations in this paper. I don't think it should be necessary to perform any

additional simulations to publish this work, but I think it merits mentioning that if you substantially increase the amount of cholesterol in the membranes used for both simulation and experiment, there may be some dampening of the Env tilt angles. In fact, the proposal that HIV-1 buds from lipid rafts rich in cholesterol might even be bolstered by this work--perhaps this helps them reduce tilting slightly so they are less prone to antibody neutralization? Might be an interesting discussion point and/or suggestion for a future study that doses cholesterol and looks at the effect on tilt angle for antibody binding. The manuscript in question is Brugger et al. (2006) PNAS 103:2641-2646 and I do not know the authors.

3) For the 5 CG-MD simulations performed with unliganded + liganded gp145, could the authors briefly mention if they used different random seeds (or not) to start/run these simulations? It is better if you did, but not disastrous if you did not--it is just appropriate to mention this so that the reader can make an objective assessment of the sampling situation.

Tyler Reddy, Los Alamos National Laboratory (my views do not represent those of my employer)

Reviewer #1:

Using cryo-EM studies of HIV envelope trimers embedded in lipid nano-discs, the authors provide novel and important observations about the molecular dynamics of the membrane proximal external regions (MPER) of the envelope and its accessibility to broadly neutralizing antibodies. There is a clear discussion of the relationship between this observed structures and previously published models. The findings are likely to be of high interest to the field. I have only two minor suggestions:

We thank the reviewer for the kind words.

1) Could the authors please comment of the entropic and enthalpic costs of the membrane extraction of buried parts of the envelope sequence?

In response to the reviewer's request, we now added the following discussion: "Prior direct measurement of 4E10 Fab binding to MPER peptide on HIV-1 virus mimetic liposomes by isothermal titration calorimetry revealed an enthalpy change of -25 kcal/mol.¹⁴ That exothermic process in association with a weak binding constant of 1.0 μ M Kd as determined by surface plasmon resonance suggests a significant entropic penalty. While we currently have no direct energetic measurements for the extraction of the gp41 TM by 4E10 in the nanodisc context, single-molecule atomic-force microscopy methods reveal that even multi-pass transmembrane proteins can be unfolded and extracted from the membrane at forces (pN) on par with or below those mediated by adhesion molecules.²⁸"

2) Could the authors please discuss the issue of steric hindrance in the binding of intact antibody molecules relative to Fab fragments, particularly in cases where more than a single antibody binds?

This is an insightful question. Fab fragments (~7 nm in size) will be able to access the tight MPER crawl space more readily than intact IgGs (~15 nm in size). scFv fragments (~3.5 nm in size) are even better yet, along with engineered constructs permitting greater VHVL domain flexibility relative to other domains in the immunoglobulin molecule (see Klein *et al.* (2009) *PNAS* **106**: 7385-7390). With respect to neutralization, binding of just one antibody to a trimeric spike is adequate (see Ref 34), so additional steric hindrance may not be germane to the protection process. Notwithstanding, the situation becomes even more interesting with respect to physiological humoral protection mechanisms when one compares IgG1 with IgG3 antibodies since the former has a short hinge segment (between the CH1 of the Fab and the Fc CH2CH3 module) relative to the long hinge of the latter Ig subclass. Based on engineered constructs, one could imagine that IgG3 flexibility, allowing the Fab arm to function in a more autonomous manner given the relaxed structural constraints, must have significant utility for neutralization involving the MPER. Unsurprisingly, a preponderance of anti-MPER neutralizing antibodies are of the IgG3 subclass. Efforts are currently underway to address this issue in detail by MD simulations and other approaches but such data are beyond the scope of the present work.

Reviewer #2:

This is an important study in which the structure of nanodisc-embedded spike protein was solved by cryo-electron microscopy and investigated with molecular-dynamics simulations. Structures of spike proteins with bound 4E10 bnAb Fabs revealed that the antibody binds exposed MPER, thereby altering MPER dynamics, modifying average ectodomain tilt, and imposing strain on the viral membrane and the spike's transmembrane segments, resulting in the abrogation of membrane fusion and informing future vaccine development. The manuscript is well written and clearly presented. There are only a few minor complaints.

We are grateful to the reviewer for the positive assessment of our work

When describing the nanodisc-embedded gp145 at an overall resolution of 3.9 Å it is stated that this resolution was sufficient to build an atomic model for the entire ectodomain, which seems overstated. Even though the model is described well in Methods, the map quality including the side chain densities should also be addressed briefly in Results.

We agree with the reviewer and removed the statement “This resolution was sufficient to build an atomic model for the entire ectodomain.” Instead we now write: “This map allowed us to build an atomic model for ectodomain residues 31-662 and a backbone model for residues 663-684 (see Methods for more detail).”

Figure 6. From the aligned Fabs it cannot be seen that 4E10 has the biggest lipid-interaction area.

The reviewer is correct and we apologize for the imprecise statement. In fact, 4E10 does not have greater lipid binding because its lipid-interaction area is bigger than that of the other bnAbs but because it is more hydrophobic (Refs 14 and 27). We therefore changed the statement in Figure 6 caption to “Note that the lipid-interaction surface of 4E10 is more hydrophobic than that of the other bnAbs.²⁶”

In Methods: ... parameters were determined with CTFFIND4 (ref 45)...

We are not certain what the reviewer would like us to change, but we substituted “determined” with “estimated”.

Reviewer #3:

The manuscript was well-written, and the tilting-related epitope accessibility mechanism seems compelling. My primary assigned task was to review the molecular dynamics (MD) simulation component of the manuscript, so I will add a few suggestions to the authors below.

We thank the reviewer for the suggestions concerning our MD simulations.

1) For the reader interested in the biophysics of HIV-1 Env, I'd suggest that the authors briefly mention the approximate lipid composition of the experimental nanodiscs and simulation

membranes prior to the Methods section, it was the first question that popped into my mind as I was reading about the tilt angles.

The reviewer makes a good point and so make the following statements in the Results section: “After purification in dodecyl maltoside (DDM) using the PG9 Fab for affinity chromatography followed by size-exclusion chromatography (SEC) (Figure S1a-b), the recombinant gp145 was reconstituted into nanodiscs with a molar 1.5: 1: 1.07 mixture of palmitoyl oleyl phosphatidylcholine (POPC), palmitoyl oleyl phosphatidylglycerol (POPG) and brain polar lipid extract, that is similar in composition to that of the HIV-1 membrane.²⁰” For molecular dynamics simulations, the “gp145 was embedded into a POPC bilayer containing 20% cholesterol, since the HIV-1 membrane is known to be rich in cholesterol.²⁰”

2) I think it would be worth mentioning in the manuscript that the lipidome of the HIV-1 envelope has been published, and is quite a bit richer in cholesterol than the composition used for either the experiments or the simulations in this paper. I don't think it should be necessary to perform any additional simulations to publish this work, but I think it merits mentioning that if you substantially increase the amount of cholesterol in the membranes used for both simulation and experiment, there may be some dampening of the Env tilt angles. In fact, the proposal that HIV-1 buds from lipid rafts rich in cholesterol might even be bolstered by this work--perhaps this helps them reduce tilting slightly so they are less prone to antibody neutralization? Might be an interesting discussion point and/or suggestion for a future study that doses cholesterol and looks at the effect on tilt angle for antibody binding. The manuscript in question is Brugger et al. (2006) PNAS 103:2641-2646 and I do not know the authors.

The reviewer raises an interesting hypothesis. So far, however, we have not seen convincing evidence that cholesterol does indeed influence the tilt angle of the ectodomain. When testing out conditions for MD simulations of membrane-embedded gp145, we tried both simulations in a pure POPC membrane and simulations in a 4:1 POPC:cholesterol membrane. While cholesterol had a stabilizing effect on the gp145 transmembrane domain, the reason we chose this membrane for the simulations, the membrane composition did not appear to affect the tilting behavior of the ectodomain. However, we only assessed the tilting behavior visually and did not quantify it. Based on the reviewer's hypothesis, we will now repeat simulations using membranes with different cholesterol content and quantitate the tilting behavior of the ectodomain. While we are excited to perform these experiments, we rather not speculate on the outcome at this point and therefore prefer not to discuss the possible effect of the membrane cholesterol content on the tilting behavior of the ectodomain in the current manuscript.

We indeed cited the paper by Brügger et al. (Ref 20) to explain the lipid composition we used to assemble the nanodiscs. However, we did not include cholesterol in the lipid mixture we used to assemble the nanodiscs.

3) For the 5 CG-MD simulations performed with unliganded + liganded gp145, could the authors briefly mention if they used different random seeds (or not) to start/run these simulations? It is better if you did, but not disastrous if you did not--it is just appropriate to mention this so that the reader can make an objective assessment of the sampling situation.

We thank the reviewer for pointing this out. We now state in the Methods section on the CGMD simulations that “Five simulations, starting from unique random seeds, were performed ...”.

Reviewers' Comments:

Reviewer #1:

Remarks to the Author:

The authors have addressed all the comments raised in the review.

Reviewer #3:

Remarks to the Author:

Everything looks good.

My final (unclear) comment was this:

"In Methods: ... parameters were determined with CTFFIND4 (ref 45)..."

this trivial comment was intended for you to remove "ref"

Reviewer #4:

Remarks to the Author:

The authors added a sentence to the Results section indicating that the nanodiscs were similar in lipid composition to the published HIV-1 lipidome; however, the experimental nanodiscs did not contain quantified cholesterol as the authors state even in their response, so I'm confused as to the basis for this claim since the published lipidome is majority cholesterol.

I think it is necessary to indicate with a sentence in the manuscript that the experimental nanodiscs do not match the published lipidome of the HIV-1 virion, in large part because they lack cholesterol.

There's no need to speculate that this may affect the tilt angle, but with the lack of cholesterol there's really no strong match in membrane composition here. The brain lipid extract used also contains unknown lipid components. I don't think this push back is too strong--just asking for a sentence to that effect in a reasonably prominent location so the reader is aware of the lipid composition discrepancy between the nanodisc and the genuine virion.

Maybe a sentence or two like this: "Notably, the nanodiscs did not contain cholesterol, which is the most abundant component of the HIV-1 lipidome. Nonetheless, our MD simulations with 20 % cholesterol exhibited reasonable tilt angle agreement with experiment."

Reviewer #1:

The authors have addressed all the comments raised in the review.

We are happy that we were able to satisfactorily address the reviewer's concerns.

Reviewer #2:

Everything looks good.

My final (unclear) comment was this:

*"In Methods: ... parameters were determined with CTFFIND4 (ref 45)..."
this trivial comment was intended for you to remove "ref"*

In the text, we cite reference 45 for the program CTFFIND4. In the usual style, this would be "CTFFIND4⁴⁵". Since the reference number may be confused as being part of the program name, we chose to change it to "CTFFIND4 (ref 45)", which we did whenever a reference citation followed a number. However, we assume that as long as the typesetter will understand our intention, they will change the format according to journal style.

Reviewer #3:

The authors added a sentence to the Results section indicating that the nanodiscs were similar in lipid composition to the published HIV-1 lipidome; however, the experimental nanodiscs did not contain quantified cholesterol as the authors state even in their response, so I'm confused as to the basis for this claim since the published lipidome is majority cholesterol.

I think it is necessary to indicate with a sentence in the manuscript that the experimental nanodiscs do not match the published lipidome of the HIV-1 virion, in large part because they lack cholesterol.

There's no need to speculate that this may affect the tilt angle, but with the lack of cholesterol there's really no strong match in membrane composition here. The brain lipid extract used also contains unknown lipid components. I don't think this push back is too strong--just asking for a sentence to that effect in a reasonably prominent location so the reader is aware of the lipid composition discrepancy between the nanodisc and the genuine virion.

Maybe a sentence or two like this: "Notably, the nanodiscs did not contain cholesterol, which is the most abundant component of the HIV-1 lipidome. Nonetheless, our MD simulations with 20 % cholesterol exhibited reasonable tilt angle agreement with experiment."

We thank the reviewer for suggesting this solution. We have now added the following statement to the section, in which we describe the reconstitution of gp145 into nanodiscs:

"This lipid mixture is similar in composition to that of the HIV-1 membrane,²⁰ except that it is missing cholesterol. Even though cholesterol is the most abundant component of the HIV-1

lipidome, our CGMD simulations performed with a membrane containing 20% cholesterol exhibited dynamic behavior of the ectodomain, in particular tilt angles, consistent with those seen in our cryo-EM maps (see below).”